# Tele-Catch: Adaptive Teleoperation for Dexterous Dynamic 3D Object Catching

## Abstract

Teleoperation is a key paradigm for transferring human dexterity to robots, yet most prior work targets objects that are initially static, such as grasping or manipulation. Dynamic object catch, where objects move before contact, remains underexplored. Pure teleoperation in this task often fails due to timing, pose, and force errors, highlighting the need for shared autonomy that combines human input with autonomous policies. To this end, we present Tele-Catch, a systematic framework for dexterous hand teleoperation in dynamic object catching. At its core, we design DAIM, a dynamics-aware adaptive integration mechanism that realizes shared autonomy by fusing glove-based teleoperation signals into the diffusion policy denoising process. It adaptively modulates control based on the interaction object state. To improve policy robustness, we introduce DP-U3R, which integrates unsupervised geometric representations from point cloud observations into diffusion policy learning, enabling geometry-aware decision making. Extensive experiments demonstrate that Tele-Catch significantly improves accuracy and robustness in dynamic catching tasks, while also exhibiting strong generalization across distinct dexterous hand embodiments and previously unseen object categories. Demonstration videos are provided in the supplementary material.

## 1 Introduction

Robotic teleoperation is a fundamental paradigm for transferring human dexterity to robotic systems (Honerkamp et al., 2025; He et al., 2024; Fu et al., 2024; Hagita et al., 2024), enabling applications in human–robot collaboration, remote operation, and industrial automation. Most existing research has concentrated on manipulating objects that are initially static, including grasping, manipulation, and in-hand adjustment (Huang et al., 2025; Li et al., 2023; Yin et al., 2025b). In contrast, dynamic object interaction, in particular the real-time catching of free-falling or moving objects, remains underexplored despite its importance, as such capability is frequently required in human–object interaction and manipulation tasks. In this work, we conduct systematic studies of multi-fingered dexterous hand teleoperation for dynamic object catching.

Despite its promise, pure teleoperation (based solely on retargeting) for dynamic object catching faces several fundamental challenges, as summarized in Fig. 1. (1) Interaction timing is highly sensitive in dynamic scenarios, where even slight delays or premature actions frequently lead to failure. (2) Interaction pose estimation remains challenging, as rapid object movement hinder the reliable perception of feasible contact configurations. (3) Interaction force regulation is difficult, since insufficient force often results in slippage, while excessive force destabilizes the manipulation. (4) Retargeting errors arise from the structural mismatch between human hands and robotic hands, which become more pronounced under fast motions (Yin et al., 2025a). To this end, we turn to learning-based frameworks as a potential foundation for reliable control. Reinforcement Learning (RL) (Lan et al., 2024) and Diffusion Policy (DP) (Chi et al., 2023; Ze et al., 2024) demonstrate strong capabilities in acquiring stable dynamic manipulation strategies, while teleoperation provides adaptability and flexibility. This motivates the inquiry into whether teleoperation inputs can be incorporated into learned frameworks to ensure robustness and stability in dynamic object catching.

In this regard, we introduce Tele-Catch, a shared-autonomy framework for dexterous hand teleoperation in dynamic object catching. The core of our shared autonomy is DAIM, a **d**ynamics-aware **a**daptive **i**ntegration **m**echanism that fuses glove-based teleoperation inputs into the diffusion de-

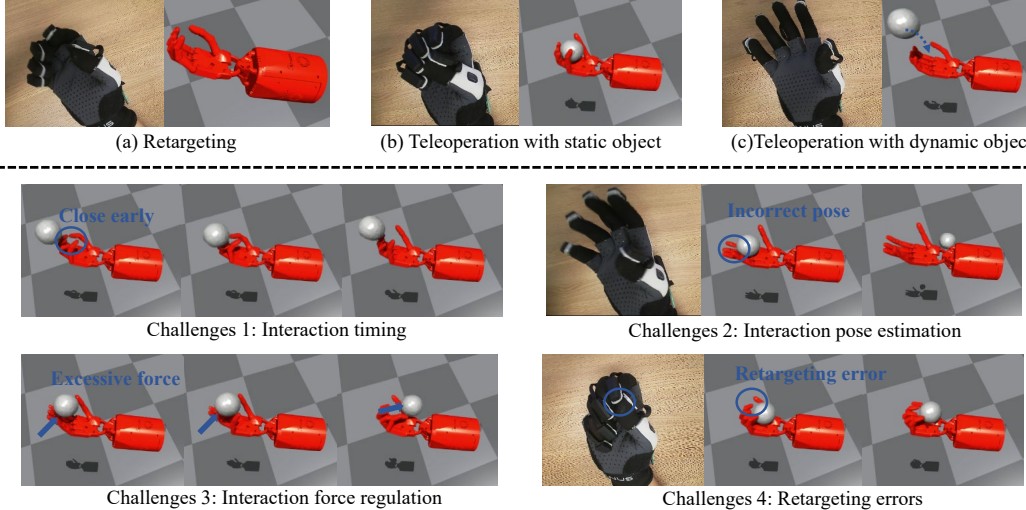

Figure 1: Teleoperation in Dexterous Manipulation. (a) Retargeting maps human hand motions to robotic hands for teleoperation. (b) Teleoperation with static object interaction. (c) Teleoperation with dynamic object interaction (our focus). Four representative cases illustrate the key challenges in teleoperation with dynamic objects: (1) grasping too early, (2) incorrect grasp pose, (3) excessive grasping force, and (4) retargeting errors between human and robotic hands, all of which eventually cause the object to slip or fall.

noising steps while adapting the integration strength to the object's dynamic state. In particular, the cosine schedule used in DAIM allows the teleoperation influence to increase smoothly over diffusion steps, avoiding abrupt changes and ensuring stable shared autonomy. This design yields a smooth and flexible control scheme that balances robustness from the learned policy with adaptability from teleoperation input.

To further enhance policy robustness and generalization, we develop DP-U3R, which augments the diffusion policy with unsupervised 3D point-cloud representations. Specifically, point-cloud observations are perturbed with Gaussian noise and processed by an encoder to extract both point-wise geometric features and pooled global structure, which are combined through attention to yield geometry-aware representations. To save the calculation cost, only global structure features are integrated into the observation space of the diffusion policy, enabling it to leverage geometric priors for more accurate action generation. Extensive experiments on mainstream data and robotic platforms validate the effectiveness and generalization of our method, showing substantial improvements in teleoperated dynamic object catching. The contributions of our work are as follows:

- We introduce Tele-Catch, pioneering the systematic study of dexterous hand teleoperation for dynamic object catching and opening a new direction beyond static grasping tasks.
- We design a dynamics-aware adaptive integration mechanism that smoothly blends glove teleoperation with learned policies, improving stability under diverse motion conditions.
- We develop a concise yet effective modular DP-U3R, which incorporates unsupervised point-cloud geometric representations into the diffusion policy, enhancing its robustness and generalization.
- Experiments conducted on representative data and hardware platforms confirm the robustness and transferability of our approach, yielding notable gains in dynamic object catching under teleoperation.

## 2 RELATED WORK

**Teleoperation Interfaces and Control.** Teleoperation is an important paradigm for transferring human dexterity to robotic systems, and current approaches fall into motion-based (Xia et al., 2024; Zhang et al., 2025), vision-based (Handa et al., 2020; Li et al., 2022; Cheng et al., 2025), and

exoskeleton-based (Chao et al., 2025; Dong et al., 2025) categories. For instance, Doglove (Zhang et al., 2025) primarily focuses on achieving dexterous hand control using low-cost materials, without incorporating additional visual systems or learning-based policies. In addition, ByteDexter-Teleop (Wen et al., 2025) employs the Manus glove with optimization-based motion retargeting to control the 20-dof dexterous hand, enabling real-time dexterous in-hand manipulation and long-horizon tasks in real-world teleoperation. Furthermore, DexterityGen (Yin et al., 2025b) incorporates external teleoperation weights into the diffusion policy denoising process, but the fixed hard threshold hinders an effective trade-off between autonomous and teleoperated control. Neglecting visual inputs limits the method's robustness and scalability.

In contrast, our work proposes a dynamics-aware adaptive integration mechanism that achieves smooth and stable coordination between glove-based teleoperation and learned policies, with 3D point cloud visual input further contributing to enhanced stability under varying motion conditions. On the other hand, most prior work (Wen et al., 2025; Yin et al., 2025b) still focuses on static object manipulation, and very few studies mention teleoperation with dynamic objects, such as catching or handover. To fill this gap, we introduce Tele-Catch, a systematic study of dexterous hand teleoperation for dynamic object catching.

**Dexterous Hand Manipulation.** Recent research on dexterous hand manipulation can be categorized into control-driven, learning-driven, and human-in-the-loop. Control-driven approaches (Jiang et al., 2024; Hess et al., 2024) rely on explicit physical models and advanced controllers such as model predictive control to manage multi-contact dynamics, enabling stable in-hand manipulation in both simulation and real-world tests. Learning-driven approaches (Yuan et al., 2025; Li et al., 2025) leverage data-driven strategies, including RL, imitation learning, and diffusion-based generative models, achieving robust dexterity and improved generalization across diverse manipulation tasks. Finally, human-in-the-loop approaches (Wang et al., 2024; Liu et al., 2025) integrate human guidance through teleoperation interfaces—such as motion-capture gloves, vision-based tracking, or exoskeleton devices—providing intuitive control of dexterous hands and generating high-quality demonstrations that enhance adaptation for complex manipulation scenarios. In this work, we attempt to optimize the learning policy to assist and enhance teleoperation.

**Point Cloud Representation for Embodied AI.** Recent works (Chen et al., 2025b; Yang et al., 2025) recognize that 3D point clouds, with their inherent state representations, effectively improve multi-fingered hand manipulation and teleoperation. Specifically, DP3 (Ze et al., 2024) develops a visual imitation learning algorithm that integrates 3D visual representations with diffusion policies. It effectively handles various robotic tasks and achieves strong generalization with few demonstrations. In addition, ViViDex (Chen et al., 2025a) proposes a vision-based framework that transforms depth-based point clouds into hand-centric coordinates to better capture hand–object geometry, enabling robust visual policies for multi-finger control without relying on privileged object states. In human-in-the-loop settings, teleoperation systems have begun to incorporate point clouds for state estimation and policy learning – for example, CordViP (Fu et al., 2025) utilizes accurate object pose and robot proprioception to produce interaction-aware point clouds of the hand and object, and pre-trains an observation encoder on hand–object contact correspondences to guide a diffusion policy. In contrast, our work seeks to obtain self-supervised geometric features from dynamic object point clouds and incorporate them into the learning policy to enhance the system robustness.

## 3 PRELIMINARIES

**Reinforcement Learning (RL).** RL provides a framework for sequential decision-making, where an agent learns to maximize long-term returns through interaction with the environment Chen et al. (2022). At each timestep $t$, the agent observes a state $s_t$, samples an action $a_t \sim \pi_\theta(a_t|s_t)$, receives a reward $r_t$, and transitions to the next state $s_{t+1}$. The optimization objective is the expected discounted return $J(\theta) = \mathbb{E}_{\pi_\theta}\left[\sum_{t=0}^{T} \gamma^t r_t\right]$, where $\gamma \in (0,1)$ denotes the discount factor. Moreover, Proximal Policy Optimization (PPO) (Schulman et al., 2017) is a widely used algorithm for continuous control, which improves upon standard policy gradients with a clipped surrogate objective that stabilizes training by constraining policy updates. PPO updates the RL policy by the clipped surrogate loss:

$$L^{\text{CLIP}}(\theta) = \mathbb{E}_t\left[\min\left(r_t(\theta)\hat{A}_t,\ \text{clip}(r_t(\theta), 1-\epsilon, 1+\epsilon)\hat{A}_t\right)\right], \tag{1}$$

with probability ratio $r_t(\theta) = \frac{\pi_\theta(a_t|s_t)}{\pi_{\theta_{\text{old}}}(a_t|s_t)}$ and advantage estimate $\hat{A}_t$. The reward function encodes task-specific objectives, thereby shaping the learned policy behavior.

**Diffusion Policy (DP).** DP (Chi et al., 2023) is built upon the framework of the denoising diffusion probabilistic model (DDPM) (Ho et al., 2020), where the diffusion mechanism is applied to action trajectories instead of raw data. DDPM processes data by gradually adding Gaussian noise and training a neural network to reverse this process. In inference, a sample is drawn from pure noise and iteratively denoised. DP adapts this principle to action trajectories, enabling multimodal and temporally coherent control generation. Formally, given a clean sample $x_0$, the forward diffusion process produces noisy versions $x_k$:

$$q(x_k \mid x_{k-1}) = \mathcal{N}\left(\sqrt{1 - \beta_k}\, x_{k-1},\, \beta_k I\right), \tag{2}$$

where $\beta_k$ is the variance schedule at step $k$. After sufficient steps ($k = K$), the data becomes nearly Gaussian noise. The reverse process, parameterized by a neural network $\epsilon_\theta$, predicts either the clean data or the added noise:

$$p_\theta(x_{k-1} \mid x_k) = \mathcal{N}(\mu_\theta(x_k, k),\, \Sigma_\theta). \tag{3}$$

By iteratively denoising from noise $x_K \sim \mathcal{N}(0, I)$, DDPMs recover samples consistent with the training distribution. This generative mechanism provides the foundation for DP.

## 4 METHOD

### 4.1 PROBLEM STATEMENT

We investigate dynamic object catching, the problem of enabling a dexterous robotic hand, controlled through a real-world teleoperation glove, to catch moving objects within a simulated environment. Let the state and point cloud at time $t$ be $s_t \in \mathcal{S}$ and $p_t \in \mathcal{P}$, the glove input $h_t \in \mathcal{H}$, and the hand action $a_t \in \mathcal{A}$, with transitions governed by $s_{t+1} \sim P(s_{t+1} \mid s_t, a_t)$, which encode objectives such as successful catching and stability. The objective of Tele-Catch is to learn a shared-autonomy policy $\pi_\theta(a_t \mid s_t, p_t, h_t)$ that enables a dexterous robotic hand to stably catch free-falling or dynamically moving objects. Given the environment state $s_t$, point cloud $p_t$, glove input $h_t$, and hand action $a_t$, the policy is trained to ensure robustness and adaptability under dynamic conditions.

Figure 2: Our Method. For clarity, we present our framework by dividing it into four components: (a) Train RL Policy (b) Data Collection (c) Train DP-U3R (d) Teleoperation. Recon. stands for Reconstruction and Seq. is Sequence.

## 4.2 Overview

As shown in Fig. 2, our framework consists of four stages. (a) We first train an RL policy in simulation to acquire stable dynamic catching skills. (b) The trained policy is then adopted to collect large-scale trajectories of state–action pairs and point cloud observations, which are filtered by the designed quality evaluation. (c) These collected data are used to train our DP-U3R, where point clouds are perturbed, encoded, and fused via pooling and attention to provide geometry-aware features that enhance the diffusion policy. (d) During teleoperation, we develop the DAIM that injects glove signals into the denoising process, adaptively balancing DP-derived policy stability with human guidance based on object dynamics and diffusion steps.

## 4.3 Train RL Policy

In the first stage, we employ the PPO algorithm to train an RL policy for dynamic object catching. PPO is selected due to its stability and effectiveness in high-dimensional continuous control, making it well-suited for dexterous hand manipulation tasks. To guide learning, we design a composite reward function tailored to the catching task, which integrates distance, orientation, and fingertip–object contact terms, along with penalties on fingertip velocities, action magnitude, torque, and energy consumption. A one-time drop penalty is further introduced for catastrophic failure. The complete reward is defined as a weighted sum:

$$
\begin{aligned}
R = {} & \lambda_{\text{dist}}\, R_{\text{dist}} + \lambda_{\text{rot}}\, R_{\text{rot}} + \lambda_{\text{ftip-dist}}\, R_{\text{ftip-dist}} \\
& - \lambda_{\text{lin}}\, P_{\text{ftip-linvel}} - \lambda_{\text{ang}}\, P_{\text{ftip-angvel}} \\
& - \lambda_{\text{act}}\, P_{\text{action}} - \lambda_{\text{torque}}\, P_{\text{torque}} - \lambda_{\text{power}}\, P_{\text{power}} - \lambda_{\text{drop}}\, P_{\text{drop}} \,,
\end{aligned}
\tag{4}
$$

where $R_{\text{dist}}, R_{\text{rot}}, R_{\text{ftip-dist}}$ represent distance, orientation alignment, and fingertip–object distance rewards, respectively. The penalty terms $P_{\text{ftip-linvel}}, P_{\text{ftip-angvel}}, P_{\text{action}}, P_{\text{torque}}, P_{\text{power}}$ regulate motion smoothness, torque load, and energy efficiency, while $P_{\text{drop}}$ penalizes object fall. Each component is weighted by its coefficient $\lambda$, which specifies its relative importance. The exact numerical values of all coefficients are summarized in Appendix. A.1.

The fingertip–object distance reward encourages the robotic hand to reduce the spatial gap between its fingertips and the target, thereby accelerating the convergence toward feasible grasp configurations. This guidance not only expedites the catching process but also improves grasp reliability by ensuring timely contact with the moving object. In addition, the fingertip velocity penalties regulate the linear and angular speeds of the fingers, discouraging excessively aggressive motions. By enforcing smoother trajectories, these penalties mitigate overshooting and reduce slippage risks, thereby promoting stable and physically plausible catches. Our reward design enables PPO to learn policies that achieve high catching success but also maintain efficiency and physical plausibility.

## 4.4 Data Collection

Moreover, we construct a dataset of dynamic catching trajectories using the trained RL policy. Let the environment state at time $t$ be $s_t \in \mathcal{S}$, which contains the object pose, velocity, and proprioceptive information of the robotic hand. A depth camera is mounted in the environment to provide a point cloud observation $p_t \in \mathcal{P}$, representing the 3D geometry of the scene at time $t$. The hand action is denoted by $a_t \in \mathcal{A}$, corresponding to the joint commands executed by the dexterous hand. Thus, each trajectory can be formalized as a sequence

$$
\tau = \{(s_t, p_t, a_t)\}_{t=0}^{T}.
\tag{5}
$$

To ensure the quality of the collected data, we define a quality evaluation function $Q(\tau)$ for each trajectory. A trajectory is accepted only if it satisfies two conditions: (i) the object is not dropped during the interaction, and (ii) the final object–palm distance is below a predefined threshold $\delta$. Formally,

$$
Q(\tau) = \mathbf{1}\Big[\text{no\_drop}(\tau) \,\wedge\, \|p_{\text{T}}^{obj} - p_{\text{T}}^{palm}\|_2 < \delta\Big],
\tag{6}
$$

where $p_{\text{T}}^{obj}$ and $p_{\text{T}}^{palm}$ denote the positions of the object and the palm center at the final timestep $T$. Only trajectories with $Q(\tau) = 1$ are retained and stored, while the rest are discarded. This

quality-controlled process produces a dataset of reliable state–action–point cloud sequences, which subsequently serves as the foundation for training diffusion-based policies.

## 4.5 DIFFUSION POLICY WITH UNSUPERVISED 3D REPRESENTATION (DP-U3R)

Additionally, we train our proposed DP-U3R. Let the (state, point cloud, action) sequence collected from RL be denoted as $\tau_s = \{(s_t, p_t, a_t)\}_{t=0}^T$. Each point cloud $p_t$ is perturbed with Gaussian noise $\xi_t \sim \mathcal{N}(0, \sigma^2 I)$ to encourage robust feature learning, yielding $\tilde{p}_t = p_t + \xi_t$. The perturbed point cloud is encoded by an encoder $E(\cdot)$, producing per-point features $f_{i,t} \in \mathbb{R}^d$. A global feature vector $f_t^g \in \mathbb{R}^d$ is then obtained by a pooling operator $P(\cdot)$:

$$f_{i,t} = E(\tilde{p}_t); f_t^g = P(\{f_{i,t}\}_{i=1}^{M_p}), \tag{7}$$

where $M_p$ is the number of points in the cloud. An attention module $A(\cdot)$ fuses local features with the global representation, yielding the final embedding $z_t \in \mathbb{R}^d$:

$$z_t = A(\{f_{i,t}\}_{i=1}^{M_p}, f_t^g). \tag{8}$$

The embedding feature $z_t$ will be used to predict and reconstruct the point cloud $\hat{p}_t$. Furthermore, the diffusion policy receives the augmented state $\tilde{s}_t = [s_t, f_t^g]$ along with the action sequence to model the denoising process. DP-U3R is optimized by two complementary objectives. The first is the point cloud reconstruction loss, defined as the L2 distance between the original point cloud $p_t$ and its reconstruction $\hat{p}_t$:

$$\mathcal{L}_{\text{recon}} = \frac{1}{M_p} \sum_{i=1}^{M_p} \|p_{i,t} - \hat{p}_{i,t}\|_2^2. \tag{9}$$

The second is the noise prediction loss for the diffusion model. At each step $t$, the policy predicts Gaussian noise $\hat{\epsilon}_t$ from the noisy action $\tilde{a}_t$, while the ground truth noise is $\epsilon_t \sim \mathcal{N}(0, I)$. The loss is given by:

$$\mathcal{L}_{\text{noise}} = \mathbb{E}_{t, \epsilon_t \sim \mathcal{N}(0,I)} \left[ \|\epsilon_t - \hat{\epsilon}_t\|_2^2 \right]. \tag{10}$$

The overall training objective is a weighted sum of the two terms:

$$\mathcal{L} = \lambda_{\text{recon}} \mathcal{L}_{\text{recon}} + \lambda_{\text{noise}} \mathcal{L}_{\text{noise}}, \tag{11}$$

where $\lambda_{\text{recon}}$ and $\lambda_{\text{noise}}$ control the balance between unsupervised point cloud learning and diffusion policy optimization.

## 4.6 DYNAMICS-AWARE ADAPTIVE INTEGRATION MECHANISM (DAIM)

In the final stage, we integrate teleoperation signals from a real glove interface into the diffusion policy via our proposed DAIM. Let the action predicted by the diffusion policy at step $k$ be $\hat{x}_k$, and the reference action derived from glove input be $x_{\text{ref}}$. The integrated action $\tilde{x}_k$ is obtained by a convex combination of the two signals:

$$\tilde{x}_k = \hat{x}_k + \alpha(k) (x_{\text{ref}} - \hat{x}_k), \tag{12}$$

where $\alpha(k) \in [0, 1]$ controls the degree of teleoperation intervention. The temporal schedule of $\alpha(k)$ is defined as:

$$\alpha(k) = \alpha_{\max} \cdot \left( 1 - \cos\left( \frac{\pi k}{2K} \right) \right), \tag{13}$$

where $K$ denotes the denoising horizon. The maximum integration weight $\alpha_{\max}$ is determined adaptively based on object dynamics:

$$\alpha_{\max} = \text{sigmoid}(u_0 - u), \tag{14}$$

where $u_0$ is the adjustment coefficient and the dynamic factor $u$ is computed as

$$u = \beta_v \cdot \frac{\|v\|}{v_0} + \beta_\omega \cdot \frac{\|\omega\|}{\omega_0}, \tag{15}$$

where $v$ and $\omega$ denote the linear and angular velocities of the object, respectively, and $v_0, \omega_0$ are normalization constants. The coefficients $\beta_v, \beta_\omega$ balance the contribution of translational and rotational dynamics. Intuitively, our DAIM adaptively adjusts the strength of teleoperation guidance according to both the diffusion step $k$ and the current object dynamics $(v, \omega)$. When the object moves rapidly, the system relies more on the diffusion policy for stability; when the motion is slower, the teleoperation input is granted more influence. This design enables stable and flexible integration of human control with learned policies in dynamic catching tasks.

## 5 EXPERIMENTS

### 5.1 EXPERIMENT SETTINGS

**Hardware and Software Settings.** As illustrated in Fig. 3, our experimental setup integrates both hardware and simulation components. On the hardware side, we employ a Manus glove(Fig. 3(a)), which captures human hand kinematics, including finger joint angles and overall hand pose, providing high-fidelity teleoperation signals. Additionally, all training and inference is conducted on one single NVIDIA RTX 4090 GPU. In simulation, we adopt Isaac Gym as the physics engine, and model the manipulator with a ShadowHand (Fig. 3(b)), a five-finger dexterous robotic hand with 20 degrees of freedom. To provide visual observations, we place an RGB-D camera in the environment, and the captured depth maps are converted into point clouds (Fig. 3(c)), which are uniformly downsampled to 1,300 points per frame. The experimental objects (Fig. 3(d)) are drawn from the public DexGraspNet dataset (Wang et al., 2023).

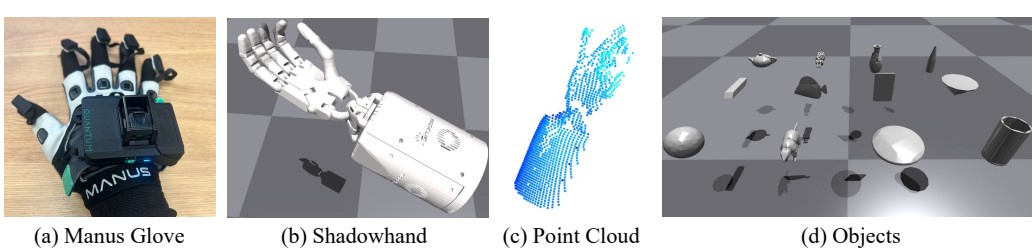

| (a) Manus Glove | (b) Shadowhand | (c) Point Cloud | (d) Objects |

Figure 3: Experiment Settings

**Implementation Details.** During RL data collection, we construct training and validation sets with a ratio of 5:1. Specifically, for each object category, we sample 5,000 successful trajectories as training data and 1,000 successful trajectories as validation data. For point cloud processing, the encoder outputs per-point features, and the global feature dimension is set to 256. The hyperparameters for DAIM are fixed as $\beta_v = 10$, $\beta_\omega = 0.1$, $u_0 = 1.0$, $v_0 = 1.0$, and $w_0 = 10$. In our DP-U3R training, the loss weights are set as $\lambda_{\text{recon}} = 1.0$ and $\lambda_{\text{noise}} = 1.0$. The detailed RL reward coefficients and their formulations are reported in Appendix. A.1. The detailed setting of the PPO and the Isaac Gym is recorded in the Tab. 5 and Tab. 6 in the Appendix, respectively.

**Evaluation Metrics.** Following prior work (Lan et al., 2024; Ze et al., 2024; Yin et al., 2025b) in dexterous manipulation, we adopt Success Rate (%) as the primary evaluation metric, where a trial is considered successful if the robotic hand catches the object and maintains stable contact for a fixed duration. For teleoperation, each object is tested 15 times, and the average success rate is reported. In addition, we measure the denoising accuracy of the diffusion policy using the Mean Squared Error (MSE) between predicted denoised actions and ground-truth actions, which directly reflects the precision of action prediction.

### 5.2 EXPERIMENT RESULTS

We report our experiment results in Tab. 1. Pure Teleoperation (Tele-Pure), which directs glove retargeting-based control, attains only 35.3% due to timing, force, and mapping errors, whereas our Tele-Catch framework improves performance to 54.7%, with notable gains on dynamic objects such as Truck and Teapot. These results confirm the effectiveness of our dynamics-aware integration and demonstrate that, although Tele-Catch advances human-in-the-loop dynamic catching, it remains a challenging task with substantial room for further research.

Table 1: Success rate comparison. Tele-Pure abbreviates Pure Teleoperation.

| Methods | Ball | Bear | Cookie | Cup | Fish | Ipod | Showerhead | Teapot | Truck | Vase | Mean↑ |
|---|---|---|---|---|---|---|---|---|---|---|---|
| Tele-Pure | 20.0 | 46.7 | **80.0** | 26.7 | 6.7 | **13.3** | 60.0 | 33.3 | 46.7 | 20.0 | 35.3 |
| Tele-Catch | **46.7** | **53.3** | **80.0** | **46.7** | **20.0** | **13.3** | **80.0** | **86.7** | **73.3** | **46.7** | **54.7** |

We provide the qualitative analysis of Tele-Catch in Fig.4, showing how glove inputs, robotic responses, and point cloud observations evolve during dynamic catching. Our method benefits from

DP-U3R priors, which provide stable guidance during unstable object motion and ensure more reliable catching. Supplementary video further illustrates this contrast, highlighting the failures of direct teleoperation and the stable performance of Tele-Catch. The supplementary video also demonstrates that our framework can effectively mitigate the four challenges identified in the introduction—grasp timing, inaccurate postures, excessive force, and mapping errors—by adaptively balancing policy stability with teleoperation guidance.

Figure 4: Qualitative analysis. The top row shows the teleoperation glove guidance, the middle row depicts the corresponding robotic hand responses in simulation, and the bottom row illustrates the synchronized point-cloud observations. Together, these visualizations demonstrate how Tele-Catch integrates human teleoperation with geometry-aware policy control to achieve stable dynamic object catching over time.

## 5.3 ABLATION STUDY

We reports the action noise MSE across different objects in Tab. 2. Overall, DP-U3R achieves the lowest mean error of 0.146, outperforming both the standard diffusion policy (0.150) and DP3 (0.148). This indicates that the incorporation of unsupervised 3D point cloud representations improves the denoising capability of the diffusion policy, leading to more accurate action predictions. On a per-object level, DP-U3R performs best or competitively across most categories. For instance, it achieves the lowest errors on Cup (0.129), Bear (0.142), and Vase (0.134). Even in categories where all methods perform similarly, such as Fish (0.136), DP-U3R maintains comparable performance. These results highlight the robustness of DP-U3R in handling diverse geometries and dynamics. Since DP3. (Ze et al., 2024) incorporates additional 3D features compared to DP (Chi et al., 2023), and DP-3UR further integrates unsupervised point cloud reconstruction features beyond DP3, Tab. 2 can also be regarded as an ablation study.

Table 2: Action noise MSE results across different objects.

|  | Ball | Bear | Cookie | Cup | Fish | Ipod | Shower. | Teapot | Truck | Vase | Mean↓ |
|---|---|---|---|---|---|---|---|---|---|---|---|
| DP | 0.152 | 0.145 | 0.158 | 0.139 | **0.136** | 0.155 | 0.156 | 0.156 | 0.161 | 0.140 | 0.150 |
| DP3 | **0.150** | 0.143 | 0.157 | 0.133 | **0.136** | 0.155 | 0.159 | **0.152** | 0.159 | 0.136 | 0.148 |
| DP-U3R | **0.150** | **0.142** | **0.155** | **0.129** | **0.136** | **0.154** | **0.154** | **0.152** | 0.158 | **0.134** | **0.146** |

## 5.4 GENERALIZATION ANALYSIS

Beyond evaluating Tele-Catch under its standard training and execution setting, we further investigate its generalization ability across different embodiments and unseen object categories. As recorded in Tab. 3, we design two complementary experiments: cross-embodied generalization, where Tele-Catch is transferred to a new dexterous hand with different kinematics, and unseen category generalization, where the method is tested on novel object categories without retraining.

**Cross-embodied generalization.** We evaluate cross-embodied performance by transferring Tele-Catch from ShadowHand to LinkerHand-L20. Since the kinematics differ, this setting requires retraining the policy. Results show that Tele-Catch consistently improves success rates compared with pure teleoperation (73.3% vs. 40.0% on Teapot), demonstrating that our framework is able to generalize across different dexterous hand embodiments. This highlights the flexibility of Tele-Catch in adapting to new robotic hardware once retrained.

**Unseen category generalization.** We further evaluate Tele-Catch on unseen object categories without retraining, including Eraser, Telephone, and Bottle. The method consistently outperforms direct teleoperation (e.g., 66.7% vs. 53.3% on Eraser), demonstrating robustness and non-trivial zero-shot generalization. These results suggest that scaling to large multi-category datasets could yield a more resilient system capable of handling common real-world objects.

Table 3: Generalization analysis.

| Objects | ShadowHand | LinkerHand-L20 | Seen | Unseen | Tele-Pure | Tele-Catch |
|---------|:----------:|:--------------:|:----:|:------:|:---------:|:----------:|
| Teapot | ✓ | | ✓ | | 33.3 | **86.7** |
| Teapot | | ✓ | ✓ | | 40.0 | **73.3** |
| Eraser | ✓ | | | ✓ | 53.3 | **66.7** |
| Telephone | ✓ | | | ✓ | 46.7 | **53.3** |
| Bottle | ✓ | | | ✓ | 20.0 | **33.3** |

## 5.5 SENSITIVITY ANALYSIS

As illustrated in Fig.5, we conduct a sensitivity analysis focusing on the ring finger position under teleoperation guidance. We press the ring finger to observe the corresponding deformation of the dexterous hand's ring finger. During the unstable phase of object motion, the robotic hand mainly relies on the diffusion policy to stabilize its response, which enables reliable catching despite rapid dynamics. Once the object is successfully secured and enters a stable state, the teleoperation input gradually becomes dominant, leading the ring finger to bend downward according to the glove command. This behavior highlights the adaptive nature of DAIM, where human guidance and the learned policy are dynamically weighted to ensure stability in catching and responsiveness in teleoperation.

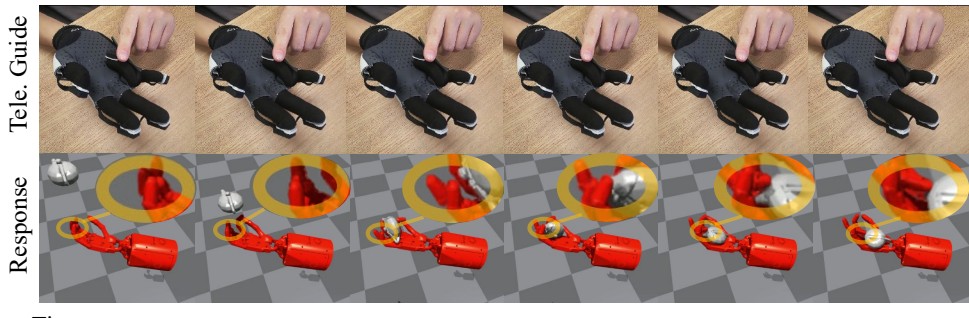

Figure 5: Sensitivity analysis. The orange ellipse highlights the variation of the ring finger.

## 6 CONCLUSION

In this work, we introduced Tele-Catch, the systematic study of dexterous hand teleoperation for dynamic object catching within a shared-autonomy framework. At its core, Tele-Catch leverages DAIM, a dynamics-aware adaptive integration mechanism that fuses glove-based teleoperation inputs into the diffusion policy's denoising process, ensuring smooth and adaptive control under varying object dynamics. To further enhance robustness and generalization, we proposed DP-U3R, which augments diffusion policy learning with unsupervised 3D point-cloud representations, enabling geometry-aware decision making. Experimental results demonstrate that Tele-Catch significantly improves success rates over baselines, achieving robust performance across diverse objects and conditions. Moreover, the analysis highlights both the effectiveness of our contributions and the remaining challenges of this new task, pointing to substantial opportunities for future research. In the future, we will attempt to explore sim-to-real transfer for deployment on real robotic hardware.

## 7 ETHICS STATEMENT

Our study does not involve human subjects, sensitive data, or applications with potential ethical risks. We confirm that this work raises no known ethical concerns.

## 8 REPRODUCIBILITY STATEMENT

We provide nearly all variables in both the main text and the appendix, including hyperparameters, model configurations, simulation settings, and hardware specifications, to ensure reproducibility. The code will be released upon paper acceptance to ensure the reproducibility and validity of the experiments and methods.

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

## A APPENDIX

### A.1 RL REWARD DESIGN DETAILS

In this section, we provide the detailed definitions of all reward and penalty terms used in PPO training. The corresponding coefficients are listed in Tab. 4.

Table 4: Reward components and coefficients used in our RL training.

| Component | Symbol | Coefficient | Coefficient Value |
|---|---|---|---|
| Hand–object distance reward | $R_{\text{dist}}$ | $\lambda_{\text{dist}}$ | 10.0 |
| Object orientation reward | $R_{\text{rot}}$ | $\lambda_{\text{rot}}$ | 0.1 |
| Fingertip–object distance reward | $R_{\text{ftip-dist}}$ | $\lambda_{\text{ftip-dist}}$ | 10.0 |
| Fingertip linear velocity penalty | $P_{\text{ftip-linvel}}$ | $\lambda_{\text{lin}}$ | 0.3 |
| Fingertip angular velocity penalty | $P_{\text{ftip-angvel}}$ | $\lambda_{\text{ang}}$ | 0.03 |
| Action penalty | $P_{\text{action}}$ | $\lambda_{\text{act}}$ | $2e^{-4}$ |
| Torque penalty | $P_{\text{torque}}$ | $\lambda_{\text{torque}}$ | $2e^{-4}$ |
| Power penalty | $P_{\text{power}}$ | $\lambda_{\text{power}}$ | $2e^{-4}$ |
| Drop penalty (one-time) | $P_{\text{drop}}$ | $\lambda_{\text{drop}}$ | 1.0 |

### A.1.1 REWARD AND PENALTY FORMULATIONS

Let $p_{\text{hand}} \in \mathbb{R}^3$ be the palm position, $p_{\text{obj}} \in \mathbb{R}^3$ the object position, $q_{\text{obj}}$ the object orientation quaternion, $q_{\text{target}}$ the desired target orientation, $p_{\text{tip},i}$ the position of the $i$-th fingertip, $v_{\text{tip},i}$ its linear velocity, $\omega_{\text{tip},i}$ its angular velocity, $a_t$ the control action at time $t$, $\tau_t$ the applied torque vector, and $\dot{\theta}_{t,j}$ the angular velocity of the $j$-th joint.

- **Hand–object distance reward:**

$$R_{\text{dist}} = -\|p_{\text{hand}} - p_{\text{obj}}\|_2$$

- **Object orientation reward:**

$$R_{\text{rot}} = -\|q_{\text{obj}} - q_{\text{target}}\|$$

- **Fingertip–object distance reward:** For $M$ fingertips,

$$R_{\text{ftip-dist}} = -\frac{1}{M} \sum_{i=1}^{M} \|p_{\text{tip},i} - p_{\text{obj}}\|_2$$

- **Fingertip linear velocity penalty:**

$$P_{\text{ftip-linvel}} = \frac{1}{M} \sum_{i=1}^{M} \|v_{\text{tip},i}\|_2$$

- **Fingertip angular velocity penalty:**

$$P_{\text{ftip-angvel}} = \frac{1}{M} \sum_{i=1}^{M} \|\omega_{\text{tip},i}\|_2$$

- **Action penalty:**

$$P_{\text{action}} = \|a_t\|_2^2$$

- **Torque penalty:**

$$P_{\text{torque}} = \|\tau_t\|_2^2$$

- **Work penalty:**

$$P_{\text{work}} = \sum_j |\tau_{t,j} \cdot \dot{\theta}_{t,j}|$$

- **Drop penalty:**

$$P_{\text{drop}} = 100 \quad \text{(applied once when object is irrecoverably lost)}$$

### A.1.2 Reward and Penalty Records

We provide the recorded reward and penalty curves during training, as illustrated in Fig. 6. The first row shows the major reward terms, including distance reward, rotation reward, and fingertip distance reward. These curves exhibit a clear upward trend and eventually stabilize, indicating that the policy successfully learns consistent catching strategies. The second row corresponds to the auxiliary penalties, such as fingertip linear and angular velocity penalties, torque and work penalties, and action penalties. Since these terms are primarily used to regularize energy consumption and finger motion smoothness, they exhibit higher variability during training. Nevertheless, their overall trends remain bounded, suggesting that the learned policy maintains stable control while avoiding excessive actuation effort.

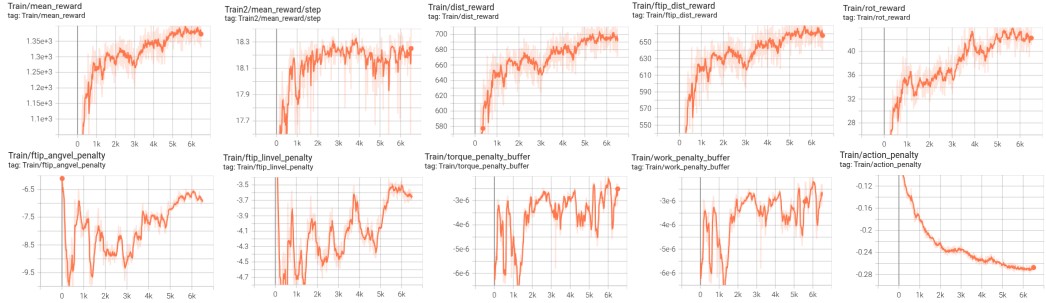

Figure 6: PPO Reward and Penalty

### A.1.3 PPO hyperparameter setting

For reproducibility, we summarize in Tab. 5 the hyperparameter configurations used in training our PPO-based RL baseline. The table includes general settings, policy network architectures, training schedules, optimization details, regularization terms, RL parameters, and logging options. These choices follow standard PPO practices with minor adjustments to ensure stable training in dynamic catching tasks.

## A.2 Simulation

### A.2.1 Simulation Setting

For completeness, we list in Tab. 6 the key simulation parameters used in Isaac Gym during RL and evaluation. The table summarizes environment configurations, control settings, initialization noise, success criteria, randomization ranges, and physics engine details. These parameters ensure that the training environment is both diverse and stable, facilitating the learning of robust catching strategies. Such randomized yet bounded settings improve generalization by preventing the policy from overfitting to a single simulation condition.

| Category | Hyperparameters |
|---|---|
| General | Random seed = 22,  Observation clipping = 5.0,  Action clipping = 1.0 |
| Policy Network | Actor hidden layers = [1024, 1024, 512] |
| | Critic hidden layers = [1024, 1024, 512],   Activation = ELU |
| Training Setup | Maximum iterations = 6500,  Schedule = adaptive,  Save interval = 1000 |
| Optimization | Rollout steps = 8,  Epochs per update = 5,  Minibatches per epoch = 4 |
| | Max gradient norm = 1,  Learning rate = $3 \times 10^{-4}$ |
| Regularization | Clipping range = 0.2,  Entropy coefficient = 0,  Target KL divergence = 0.016 |
| RL Parameters | Discount factor $\gamma = 0.96$,  GAE parameter $\lambda = 0.95$,  Initial noise std = 0.8 |
| Logging | Log interval = 1,  Console logging = True |

Table 5: PPO hyperparameter setting.

| Category | Key Parameters |
|---|---|
| Environment | numEnvs = 8192,  envSpacing = 0.75,  episodeLength = 50 |
| Control | dofSpeedScale = 20,  controlFrequency = 60 Hz |
| Noise (init/reset) | posNoise = 0.01,  dofPosRandom = 0.2 |
| Task Success | successTolerance = 0.1 |
| Randomization | hand/object mass [0.5, 1.5],  friction [0.7, 1.3],  object scale [0.95, 1.05] |
| Physics (PhysX) | solver = TGS,  numPosIter = 8,  contactOffset = 0.002,  bounceThreshold = 0.2 |

Table 6: Isaac Gym simulation parameter setting.

### A.3  PSEUDO-CODE OF PROPOSED METHOD

To provide a clear overview of our method, we present pseudo-code descriptions of the Tele-Catch pipeline. The framework consists of two complementary phases: a training phase (Alg. 1), where RL trajectories are collected and used to train our DP-U3R model, and an execution phase(Alg. 2), where the learned policy is integrated with teleoperation signals through DAIM. The pseudocode aims to clarify the data flow and algorithmic steps beyond the conceptual diagrams given in the main text.

**Training Stage.** We employ PPO to generate successful dynamic catching trajectories. These trajectories include state information, executed actions, and synchronized point cloud observations. The collected data are then used to train DP-U3R, which encodes noisy point clouds into geometric embeddings and optimizes a diffusion policy with both reconstruction and noise prediction losses.

---

**Algorithm 1** Training Stage: RL and DP-U3R

    **Inputs:** Simulator $\mathcal{E}$, PPO config, point cloud $p_t$
    **Outputs:** DP-U3R policy $\pi_\theta$
 1: **procedure** STAGE 1: RL TRAINING
 2:    Train PPO in $\mathcal{E}$ to catch dynamic objects
 3:    Collect successful trajectories $\{(s_t, a_t, p_t)\}$
 4: **end procedure**
 5: **procedure** STAGE 2: DP-U3R TRAINING
 6:    Encode noisy point cloud $p_t$ into embedding $z_t$
 7:    Train diffusion policy $\pi_\theta(s_t, z_t)$
 8:    Optimize with reconstruction loss + noise prediction loss
 9: **end procedure**

---

**Execution Stage.** During execution, the trained DP-U3R is combined with glove-based teleoperation inputs using the dynamics-aware adaptive integration mechanism (DAIM). At each control step, the policy proposes an action from the encoded point cloud and state, while glove commands are retargeted to reference actions. DAIM dynamically fuses these two signals according to object dynamics and time step, enabling stable yet responsive teleoperation for dynamic catching.

---

**Algorithm 2** Execution Stage: Teleoperation with DAIM

---

    **Inputs:** Trained policy $\pi_\theta$, glove input $h_t$, point cloud $p_t$
    **Outputs:** Executed action $\tilde{x}_t$
1: **procedure** TELEOPERATE
2:     **for** control step $t = 1 \ldots T$ **do**
3:         Observe $(s_t, p_t)$ and compute point cloud feature $f_t^g$
4:         Generate policy action $\hat{x}_t \leftarrow \pi_\theta(s_t, f_t^g)$
5:         Retarget glove input $x_{\text{ref}}$ from $h_t$
6:         Compute DAIM weight $\alpha(k)$ based on dynamics and step
7:         Fuse actions: $\tilde{x}_k = \hat{x}_k + \alpha(k)(x_{\text{ref}} - \hat{x}_k)$
8:         Execute action $\tilde{x}_K$ on the hand
9:     **end for**
10: **end procedure**

---

### A.4 LLM USAGE STATEMENT

We employed ChatGPT-5 to assist in refining the language and enhancing the readability of the manuscript. No large language models were involved in the design of experiments, data analysis, or the derivation of scientific conclusions in this work.

