# OpenReview forum: "Tele-Catch: Adaptive Teleoperation for Dexterous Dynamic 3D Object Catching"
_ICLR.cc/2026/Conference — ICLR 2026 Conference Withdrawn Submission_

### Official Review · Reviewer_3ioW · 2025-10-24

**Soundness:** 2
**Presentation:** 2
**Contribution:** 1
**Rating:** 2
**Confidence:** 5

**Summary:**

This paper presents a framework for dexterous hand teleoperation in dynamic object catching. The proposed system comprises four main stages: (1) training a reinforcement learning (RL) policy for skill acquisition and data collection, (2) collecting state–action–point cloud trajectories, (3) training a Diffusion Policy with Unsupervised 3D Representation (DP-U3R), and (4) teleoperation using a Dynamics-Aware Adaptive Integration Mechanism (DAIM). The DP-U3R leverages point-cloud observations to augment state representations for both action generation and geometric reconstruction, while the DAIM adaptively fuses teleoperation signals with policy-generated actions based on object dynamics and diffusion steps. The framework is evaluated on dynamic object-catching tasks, where a real-world teleoperation glove controls a simulated dexterous hand in Isaac Gym.

Although the paper considers a challenging and practically relevant problem in robotic teleoperation, its methodological novelty appears limited. The core contribution, DAIM, resembles a hand-crafted engineering heuristic rather than a principled formulation, and its design choices are neither thoroughly justified nor supported by sufficient ablation analysis. In particular, the blending mechanism between diffusion outputs and teleoperation signals seems overly simplistic. While the problem setting holds potential practical value, the current presentation does not meet the level of conceptual innovation typically expected for full acceptance.

**Strengths:**

- Problem Domain: The primary strength of this paper is its focus on a challenging and underexplored problem in robotics. (dynamic object interaction via teleoperation)

**Weaknesses:**

- Limited Methodological Novelty: The main concern with this paper is its limited novelty. The core framework is largely an integration of existing, well-established components (e.g., PPO for data collection, Diffusion Policy as the autonomous controller). The central contribution appears to be the DAIM, which arbitrates between human and policy actions. However, this mechanism itself feels more like a specific, hand-tuned engineering solution rather than a fundamental or generalizable new technique.
- Insufficient Rationale for Design Choices: The paper does not include a justification for the specific design of the DAIM (Section 4.6). There is no ablative study or theoretical reasoning provided to explain why the particular functions are optimal or even well-suited for this integration.

**Questions:**

- Could you augment Figure 5 with a plot showing the value of the integration coefficient over time? This would provide a better understanding of how the system works between the user and the policy.
- Rationale for Functions: As mentioned in the weaknesses, the choice of sigmoid and cosine functions in Section 4.6 is not well-justified. Could you elaborate on the specific properties of these functions that led to their selection?
- The coefficients used in the dynamic factor seem hard-coded. It seems that the optimal values for these coefficients would depend on the properties of the object being caught (e.g., its shape, mass, or velocity). Could you provide an analysis of the system's sensitivity to these coefficients?

---

### Official Review · Reviewer_g9YN · 2025-10-26

**Soundness:** 3
**Presentation:** 3
**Contribution:** 2
**Rating:** 2
**Confidence:** 4

**Summary:**

This paper proposes a dynamic teleoperation framework that first trains an RL policy in simulation, collects data, and then trains a diffusion policy to assist dynamic teleoperation. An unsupervised 3D representation loss is incorporated during training. Experiments conducted in simulation demonstrate the effectiveness of the proposed learning approach.

**Strengths:**

1. The direction of dynamic-catch is interesting, which is an under-explored direction.

**Weaknesses:**

1. The main pipeline—training an RL policy in simulation, collecting data, and training a policy that takes glove signals as input—is largely similar to Dexterity Gen. Therefore, Dexterity Gen should be considered the baseline rather than simple teleoperation.
2. The MSE metric does not provide meaningful insight when training the diffusion policy; the success rate should be included in the ablation study to better evaluate performance.
3. The paper only presents simulation experiments without any real-world validation. Given the dynamic catching setting, the sim-to-real gap can be substantial. As a result, the reported improvements may not transfer effectively to real-world scenarios, which limits the demonstrated effectiveness of the proposed method.
4. The overall technical novelty of the paper is not very strong. It resembles a system paper that builds upon Dexterity Gen with additional designs for dynamic catching and unsupervised point cloud learning. The work may be more suitable for a robotics-focused venue rather than ICLR.

**Questions:**

1. The current results show that combining teleoperation with a pretrained policy performs better than teleoperation alone. However, it is unclear how an autonomous policy—without any glove input—would perform. What would be the success rate of such a policy? If the autonomous policy achieves a higher success rate than teleoperation combined with the pretrained policy, what is the significance of integrating teleoperation?
2. Regarding DAIM, as I understand it, the lowercase k represents the environment step, while the uppercase K denotes the denoising horizon (e.g., denoising for 10 steps to obtain the final result). I am confused about how these two are combined and used in practice.

---

### Official Review · Reviewer_Gat9 · 2025-10-31

**Soundness:** 2
**Presentation:** 3
**Contribution:** 2
**Rating:** 4
**Confidence:** 4

**Summary:**

This paper introduces Tele-Catch, a framework for dynamic object catching using dexterous hands, a challenging and underexplored task compared to static manipulation. Its core contribution is a shared autonomy system that seamlessly fuses human teleoperation with an autonomous diffusion policy. This is enabled by a novel dynamics-aware mechanism (DAIM) that adaptively integrates human input, and a policy (DP-U3R) that leverages geometric point cloud representations for robustness. The key advantages are its successful tackling of the dynamic catching problem, the effective human-robot synergy it creates, and its demonstrated generalization across different hand embodiments and unseen objects.

**Strengths:**

1. The topic investigated in this paper is highly interesting and challenging.
2. The proposed method demonstrates significant insight by effectively combining the strengths of teleoperation and existing learning-based approaches.
3. The paper provides comprehensive experimental validation to verify the effectiveness of the method and its design.

**Weaknesses:**

1. Why can h_t be directly incorporated into the diffusion policy's denoising process? Could this potentially disrupt denoising, as the training data might not have encountered such a conditioned input?
2. Some information, such as the object's linear and angular velocity, is difficult to obtain in real-world scenarios. Are there any practical solutions and corresponding experiments to address this limitation?
3. How were the dynamic objects configured in the simulation? Please specify the parameter ranges used for their linear and angular velocities.
4. Comparing action noise error is not very meaningful. A more critical metric is the task success rate, particularly in comparison to baselines like DP and DP3.
5. The cross-embodiment experiments are insufficient. More extensive validation is needed.
6. The number of test objects is too limited and should be expanded.
7. The supplementary material only shows results for the teapot. More visualizations are required to effectively demonstrate the validity of the actions.

I will consider raising the score if the rebuttal of the author can address the above concerns.

**Questions:**

please see the weaknesses

**Details Of Ethics Concerns:**

No ethics concerns

---

### Official Review · Reviewer_w51r · 2025-11-01

**Soundness:** 3
**Presentation:** 3
**Contribution:** 2
**Rating:** 2
**Confidence:** 5

**Summary:**

This paper presents a semi-autonomous approach for catching a flying object via teleoperation. The core idea is to train a catching policy using reinforcement learning in simulation, which is later guided by human gestures during deployment. The proposed method is validated using a static robotic hand in simulation.

**Strengths:**

The problem addressed in this work is quite challenging. Catching a flying object with a fixed dexterous hand requires extremely precise timing and coordination, and the paper’s attempt to tackle this problem is commendable.

**Weaknesses:**

The main weakness of this paper is that the proposed problem could likely be solved with a fully automated policy, without requiring human guidance. In fact, similar catching problems have been addressed in earlier works [1]. Furthermore, the absence of real-world experiments significantly limits the paper’s practical significance and overall impact.

[1] A Dynamical System Approach for Softly Catching a Flying Object: Theory and Experiment. TRO 2016.

**Questions:**

N/A. I would like the authors to discuss more about their motivation.

I believe one way to strengthen this paper is to show real world results on catching complex, irregular objects where prior methods fail.

---

### Note · Authors · 2025-11-13

I have read and agree with the venue's withdrawal policy on behalf of myself and my co-authors.